# Extracellular Vesicles as an Efficient and Versatile System for Drug Delivery

**DOI:** 10.3390/cells9102191

**Published:** 2020-09-29

**Authors:** Xuan T. T. Dang, Jayasinghe Migara Kavishka, Daniel Xin Zhang, Marco Pirisinu, Minh T. N. Le

**Affiliations:** 1Department of Pharmacology, Yong Loo Lin School of Medicine, National University of Singapore, Singapore 117600, Singapore; phcdttx@nus.edu.sg (X.T.T.D.); migarakj@u.nus.edu (J.M.K.); 2Department of Biomedical Sciences, College of Veterinary Medicine and Life Sciences, City University of Hong Kong, Kowloon, Hong Kong; daniel.xin.zhang@my.cityu.edu.hk (D.X.Z.); mpirisin@cityu.edu.hk (M.P.); 3Department of Molecular Biology and Genetics, Cornell University, Ithaca, NY 14853-2703, USA; 4City University of Hong Kong Shenzhen Research Institute, Shenzhen 518057, Guangdong, China

**Keywords:** extracellular vesicles, drug delivery, therapeutic, clinical, translation

## Abstract

Despite the recent advances in drug development, the majority of novel therapeutics have not been successfully translated into clinical applications. One of the major factors hindering their clinical translation is the lack of a safe, non-immunogenic delivery system with high target specificity upon systemic administration. In this respect, extracellular vesicles (EVs), as natural carriers of bioactive cargo, have emerged as a promising solution and can be further modified to improve their therapeutic efficacy. In this review, we provide an overview of the biogenesis pathways, biochemical features, and isolation methods of EVs with an emphasis on their many intrinsic properties that make them desirable as drug carriers. We then describe in detail the current advances in EV therapeutics, focusing on how EVs can be engineered to achieve improved target specificity, better circulation kinetics, and efficient encapsulation of therapeutic payloads. We also identify the challenges and obstacles ahead for clinical translation and provide an outlook on the future perspective of EV-based therapeutics.

## 1. Introduction

Drug delivery systems are used to increase the bioavailability and therapeutic efficacy of conventional drugs by increasing the concentration of drugs at the site of diseased tissues [1,2,3,4,5,6] and inducing controlled drug release [7] while minimizing side effects and the need for repeated dose administration [8,9,10]. Following the advancement of nanobiotechnology, nanoparticles have been used as drug carriers to improve the delivery efficiency of conventional medicine while opening a new field of targeted delivery of therapeutic nucleic acids. However, only a handful of them have progressed to the market, three of which are liposome-encapsulated doxorubicin [11,12], paclitaxel enclosed in protein-based nanoparticles [13,14], and more recently a nanoparticles-encapsulated RNA interference-based drug for the treatment of polyneuropathy in patients with hereditary transthyretin-mediated amyloidosis [15]. Many factors limit the commercialization and clinical success of other nanoparticle-based therapeutics such as the high cost, difficulty in large scale manufacturing, low target specificity, and safety concerns in terms of particle agglomeration and immunogenicity [16,17,18]. In the field of drug delivery, naturally derived nanoparticles have been drawing increasing attention due to their potential to overcome the limitations of synthetic nanoparticles. Among naturally derived nanoparticles, extracellular vesicles (EVs) are among the most attractive candidates. EVs are cell-derived particles enclosed in phospholipid bilayers that are incapable of self-replication [19]. EVs were first identified as a means of intercellular communication in 1981 by Trams and colleagues when the group studied exfoliated membrane vesicles secreted into the supernatant of various normal and cancerous cell lines [20]. In 1983, EVs were further studied by Harding and colleagues as a mechanism for transferrin receptor modulation and recycling on reticulocytes [21]. After three decades, EVs have been found to play an important role in many physiological and pathological cellular functions, including cell-cell communication [22], host-parasite communication [23], immune modulation [24], thrombosis [25,26], vascular dysfunction [27], de novo mineralization [28], rejuvenation [29], and alteration of the tumor microenvironment [30,31,32]. Given the fact that EVs can serve as natural transporters to mediate the physiological functions of cells, EVs have gained popularity as highly efficient carriers, delivering cargo consisting of biological molecules to recipient cells, which in turn alter the target cell activities at the RNA, DNA, or protein level.

In this review, we focus on the therapeutic potential of EVs as an efficient drug delivery system. In the first part, we review the biogenesis and biochemical properties of EVs and discuss the current methodologies for EV isolation and characterization. Concurrently, we highlight the key advantages of EVs that make them suitable for drug delivery, with close comparison to alternate drug delivery systems currently in use. Next, we describe how EVs can be engineered to further their therapeutic potential and achieve efficient, targeted delivery of encapsulated payloads. Lastly, we outline the future potential of EV-based therapeutics and discuss the current challenges and obstacles facing their clinical translation.

## 2. Biogenesis and Classification of EVs

Secreted by most cell types, EVs consist of a broad range of particles generated from different biogenesis pathways, ranging from 30 nm to a few microns in size [33]. They are classified into several subtypes (Figure 1), namely exosomes, microvesicles, and apoptotic bodies, the first two of which are frequently grouped together for studies [33,34].

Exosomes are the small-size subtype of EVs associated with endosomal biogenesis pathways [35]. The formation of exosomes starts with the inward budding of early endosomal membrane which creates intraluminal vesicles [36]. The generation of these intraluminal vesicles in endosomes is known to be highly regulated by endosomal sorting complex required for transport (ESCRT) complexes. The ubiquitin-binding region of ESCRT-0 first recognizes and sequesters ubiquitinated proteins on the endosomal membrane. ESCRT-I and ESCRT-II are later recruited to interact with ESCRT-0 and promote the formation of inward budding with the sequestered complexes. ESCRT-III, together with other associated proteins such as VPS4 and VTA1, then mediate the scission of inward buds, releasing vesicles into the endosomal lumina [37]. Stuffers et al. reported that the depletion of ESCRT-0, -I, -II, and -III does not completely abolish the formation of intraluminal vesicles, indicating the existence of ESCRT-independent pathways [38]. With the maturation of early endosomes into late endosomes [39], endosomes with intraluminal vesicles are named multivesicular bodies (MVBs) and have two distant fates awaiting them: most of the MVBs move towards and then fuse with the plasma membrane, subsequently releasing the intraluminal vesicles, i.e., exosomes, into the extracellular space, while the rest of the MVBs will fuse with lysosomes, exposing the intraluminal vesicles to hydrolases for catabolism and further degradation [40]. Recent studies have shown that the maturation and fate determination of endosomes are regulated by many pathways involving ubiquitin ligase RNF26 and Rab GTPases [41,42].

Microvesicles are generally believed to be bigger in size than exosomes and are formed by the outward shedding of the cell membrane directly into the extracellular space [33]. The biogenesis of microvesicles is less investigated as compared to exosome biogenesis but plasma membrane curvature is considered to be important in the process [43]. Nabhan and colleagues proposed that arrestin domain-containing protein 1 is essential for the formation and release of microvesicles by translocating TSG101 from endosomes to the plasma membrane and causing the curvature of the plasma membrane [44]. Another model by Stachowiak and colleagues considered the importance of protein-protein crowding for curving the plasma membrane rather than protein-specific pathways as they observed that even green fluorescent protein (GFP), irrelevant to membrane curvature, can induce membrane bending when in high concentrations [45]. Extracellular Ca^2+^ levels also significantly affect microvesicle production [46]. A possible model explaining this is the participation of Ca^2+^-dependent enzymatic interactions such as aminophospholipid translocases which cause the asymmetrical rearrangement of plasma membrane phospholipids and the subsequent membrane curvature favorable for microvesicle budding [47,48]. This aspect has been applied to boost microvesicle production where researchers treat red blood cells with calcium ionophores to promote EV production for therapeutic purposes [49].

Apoptotic bodies are usually larger than microvesicles and are only generated by cells undergoing apoptosis [50,51]. During the execution phase of apoptosis, cells experience a great number of morphological changes including cell shrinkage and cytoskeleton breakup, which causes blebbing of the apoptotic bodies [52,53]. Given their distant biogenesis pathways related to apoptosis, apoptotic bodies are customarily excluded when EVs are referred to in a general context. Contrary to the traditional belief that apoptotic bodies are just random debris from dying cells, apoptotic bodies are increasingly recognized as crucial immune regulators and disease biomarkers [54,55].

At the moment, there is no general consensus on the biomarkers or isolation methods that can confidently separate each of the subtypes above due to the difficulty in defining biogenesis pathways of EVs, the overlapping in size, and the absence of specific biomarkers [56,57]. In 2018, the International Society for Extracellular Vesicles (ISEV) recommended that researchers, to define EVs, should describe the physical and biochemical characteristics of EVs, such as size, density, surface protein composition, and lipid composition, instead of using the broad terms “exosomes”, “microvesicles”, or “apoptotic bodies” [58].

In addition to the classic categorization system above, novel definitions of EV subpopulations have been recently introduced, such as oncosomes, migrasomes, and P200 EVs. Oncosomes are the cancer-derived EVs carrying oncogenic materials such as oncogenic proteins and microRNAs and function through tumor microenvironment conditioning [59,60]. Oncogenic molecules transferred by large oncosomes include integrin alpha V (ITGAV) which promotes prostate cancer aggressiveness such as adhesion and invasion via AKT activation [61], and microRNA-1227 which boosts the migration of cancer-associated fibroblasts [59]. Migrasomes are particles formed at the tips or at the intersections of retraction fibers when the cells migrate [62]. They consist of smaller vesicles and cellular contents, and are released into the extracellular space when retraction fibers break off [63]. Migrasomes are reported to regulate the clustering of dorsal forerunner cells [62,64]. P200 EVs are a novel subgroup of EVs believed to be overall smaller than exosomes, as reported by Lee and colleagues [65]. Notably, even though P200 EVs are isolated by further fractionation of total EVs using ultracentrifugation, CD81, an abundantly found EV marker of total EVs, is found in much lower levels in the P200 EVs subpopulation [65]. Furthermore, in a recent publication, Jeppesen et al. identified annexin A1 as a biomarker exclusively for classical microvesicles, while non-vesicular components are enriched in double-stranded DNA (dsDNA), micro RNAs (miRNA), and cytosolic proteins such as HSP90, HSPA13, and Argonaute proteins [66]. These features of non-vesicular components fit the description of the recently discovered nanoparticles termed “exomeres” [66,67]. However, classical exosomes, the EVs carrying CD63, CD81, and CD9 biomarkers, contain low levels of miRNA and RNA binding proteins and other proteins involving in the miRNA machinery [66].

## 3. Biochemical Properties of EVs and Their Intrinsic Advantages for Drug Delivery

The biochemical composition of EVs mainly consists of lipids, proteins, and nucleic acids [68]. Derived from cells, EVs obtain their traits based on the cargo transferred from the donor cells, which are known as their molecular signature [69,70]. From a therapeutic point of view, since EVs are broadly involved in intercellular communication, they are naturally endowed with a number of features that facilitate them in carrying out their function effectively. Chief among them is the fact that EVs are capable of safely and successfully transferring bioactive payloads (such as proteins, miRNAs, and mRNAs) to recipient cells [49]. Moreover, EVs have been shown to have a natural ‘homing mechanism’, resulting in specific patterns of uptake [71]. These aforementioned features exhibited by EVs also make them ideal natural vectors for the delivery of therapeutic payloads. In this section, we will detail how the biochemical features of EVs make them ideal as an efficient drug delivery platform.

### 3.1. EV Lipids

The main lipid contributor of EVs is their lipid bilayer membrane, whose structure resembles the plasma membrane of cells and is quite different from other single-layered lipids in the circulation such as cholesterol in blood lipoproteins [72,73]. With regard to molecular composition, EV lipids are enriched in phosphatidylserine (PS) which may influence the uptake of EVs [74,75]. EVs from different parental cells may also have different lipid profiles. For instance, EVs from non-tumorigenic cells are enriched in prenol lipids, glycerolipids, and fatty acids, while EVs from tumorigenic cells have higher levels of sphingolipids, glycerophospholipids, and sterol lipids [76]. Similarly, EVs from the bronchoalveolar lavage fluid of asthmatic patients are enriched in ceramide-phosphates, ceramides, and phosphatidylglycerol in comparison to healthy controls [77]. Even different subtypes of EVs can have distinctive lipidomic profiles. For instance, apoptotic bodies have PS translocated to the outer part of the lipid bilayer and surface molecules are oxidized [78].

#### 3.1.1. EV Lipids on Protection of Encapsulated Payloads

Functionally speaking, the lipid bilayer membrane of EVs serves as an ideal shelter for the enclosed cargo from external enzymatic degradation, thus securing the integrity and functionality of the cargo until its delivery to the target cells [79]. Therapeutic molecules are susceptible to degradation by enzymatic digestion in the circulation shortly after systemic administration. Encapsulation of these molecules within EVs protects them from the degradative effects of enzymes such as DNases or RNases. Our study, conducted using RBC-derived EVs, demonstrated that the encapsulation of RNA antisense oligonucleotides (ASOs), or Cas9 mRNA inside RBC-derived EVs protected the encapsulated RNAs from RNase-mediated degradation. Meanwhile, in the unencapsulated control, almost all the RNA was degraded [49]. In the case of very small therapeutics molecules (hydrodynamic diameter <8 nm), encapsulation inside EVs also protects them from renal clearance. Generally, nanoparticles with a hydrodynamic diameter of less than 6 nm are efficiently removed from the circulation by renal clearance. Larger molecules up to 8 nm are also capable of renal clearance, depending on the nature of the molecules [80]. Encapsulation inside EVs reduces the possibility of renal clearance as the EVs are significantly larger than the permissible size for filtration through the many layers of the glomerular capillary wall. This method of preventing renal clearance of small molecules via encapsulation inside EVs is preferred over other methods such as the conjugation with additional functional groups, which could then affect the downstream functions of the molecules.

#### 3.1.2. EV Lipids on Homing Properties

The homing property of EVs involves the propensity of EVs secreted from certain donor cells to be recognized and taken up by specific recipient cells. Lipid composition is increasingly recognized as a contributor for selective binding and internalization of EVs in addition to the extensively studied EV protein composites. As mentioned above, one of the most discussed molecules facilitating the recognition and internalization of EVs is PS. PS is a membrane phospholipid localized mostly in the inner leaflet of the bilayer [81]. The translocation of PS to the outer bilayer of the membrane is a signal to initiate the clotting cascade or a marker for the clearance of apoptotic bodies [82,83]. Macrophages, being responsible for the clean-up of cell debris after apoptosis, can recognize PS and internalize PS-positive EVs by endocytosis. This makes macrophages ideal recipient cells for PS-decorated EVs. Blocking PS on EVs by annexin-V does not alter the EVs size, but efficiently reduces EV internalization by macrophages in a dose-dependent manner [84]. The uptake of PS-rich EVs in the liver of a mouse model also reduced after the mice were treated with PS-rich liposomes, which confirmed that PS contributes to EV uptake [84]. Interestingly, substituting PS with another lipid with high negative charge such as phosphatidylglycerol gave a similar result in blocking the uptake of PS-rich EVs in immune cells, suggesting that the high negative charge of PS determines its biological function [84].

### 3.2. EV Proteins

EVs carry an abundant array of multifunctional proteins both on the surface and inside EVs [85,86]. Based on the origins and functions, EV proteins can be further divided into several subgroups: biogenesis-associated proteins, intracellular traffickers, surface tetraspanins, cytosolic proteins, adhesion-associated proteins, and cell-type-specific proteins [68,87]. Biogenesis-associated proteins are those that assist in the biogenesis process of EVs including ALIX, TSG101, and VPS4. For instance, ALIX is an ESCRT accessory protein and regulates endosomal pathways and thus EV biogenesis through syndecan–syntenin–ALIX axis and ALIX depletion substantially inhibits EV production [88,89,90]. Common intracellular traffickers found in EVs include Rab GTPases and annexins [91,92,93]. Rab GTPases regulate restricted membrane association and activation state, protein sorting, and vesicle tethering as a GTP/GDP-dependent molecular switch by recruiting Rab-type-specifc effector proteins [91]. Annexins are enriched in EVs but how they specifically participate in EV biogenesis and get sorted into EVs remains unclear even though previous studies suggest the role of annexins in endocytosis and maintaining membrane homeostasis [92,93,94]. Tetraspanins are the family of proteins with four transmembrane domains found abundant on EV surfaces such as CD63, CD9, and CD81 [95]. A recent study by Larios and colleagues demonstrated that the sorting of tetraspanins into EVs is dependent on ALIX and ESCRT-III. Tetraspanins are also essential for EV targeting and uptake, which we will detail in the later part of this review [95]. The aforementioned EV proteins are highly enriched in EVs and frequently used as EV markers for characterization. In the latest experimental guidelines ‘minimal information for studies of extracellular vesicles 2018’, ISEV recommends demonstrating the enrichment of EV proteins from multiple classes in protein-based EV characterization with a depletion tendency of putative contaminants [58]. One notable family of adhesion-associated proteins in EVs is integrins which are reported mostly in the context of tumor-derived EVs [96]. Similar to tetraspanins, integrins are located on the EV surface and proposed to be important for EV homing, which we will focus on in the following section [97]. Cell-type-specific proteins reflect the molecular origins of EVs. Red blood cell extracellular vesicles (RBCEVs) display surface glycophorin A, a well-known RBC marker while mesenchymal stem cell (MSC)-derived EVs exhibit surface CD73, CD90, and CD105 expression pattern [98,99,100]. Besides tracing the EV origin, researchers are also exploring the possibility of using cell-type-specific proteins on EVs to discriminate disease subtypes and to predict disease prognosis [101,102]. EV online databases such as Vesiclepedia [103] and ExoCarta [104] contain over thousands of EV protein entries, allowing for a conglomerate of EV protein knowledge.

#### EV Proteins on Homing Properties

Several membrane proteins contribute to the homing property of EVs, such as the tetraspanin protein family and integrin [71,105]. Endocytosis is the main mechanism of cellular uptake of EVs, and consists of four pathways: the clathrin-dependent pathway, the caveolae-dependent pathway, macropinocytosis, and phagocytosis [106,107,108]. As both donor cells and recipient cells have distinctive membrane proteins in the form of ligands and receptors, EVs isolated from different sources may have very different homing properties against different cell lines and tissues.

Tetraspanins are transmembrane proteins contributing to cell adhesion, migration, and signaling [109,110]. They can interact with themselves and other cellular proteins to form tetraspanin complexes, termed tetraspanin-enriched microdomains (TEM), which could mediate vesicular fusion and/or fission. This protein superfamily consists of 33 transmembrane proteins, of which CD63 and CD81 are frequently used as EV biomarkers [111]. Some tetraspanins such as CD9, CD82, and CD151 are present in almost every EV subtype, while other tetraspanin members are only expressed in particular cell lines and tissues, such as Tssc6 in hematopoietic cells [112], CD37 and CD53 in immune cells [113], and Tetraspanin-8 (Tspan8) in tumor cells [114]. The recruitment of tetraspanins into EVs during biogenesis brings along other adhesion receptors, creating tetraspanin webs to enhance the targeting specificity of EVs [105]. Rana and colleagues found that EVs obtained from the rat adenocarcinoma cell line were enriched in Tspan8 associated with CD49d, but not with CD49c, CD9, CD81, or CD151 [71]. Under the stimulation of phorbol-12-myristate-13-acetate (PMA), Tspan8 gained higher affinity to CD49d, which led to an increase in CD49d recruitment to the membrane [71]. The association of Tspan8-CD49d at the membrane during EV biogenesis brings about an enrichment of this tetraspanin complex within secreted EVs, and subsequently changes EV uptake by endothelial cells [71]. Tspan8, when associating with different proteins, can direct EVs to different cells and organs [105]. In a study on EVs isolated from adenocarcinoma cells, Rana and colleagues showed that EVs carrying Tspan8 in complex with beta4 protein has a distinctive targeting pattern compared to Tspan8 wildtype and Tspan8 with a loss-of-function modification at *N*-terminal domain [105]. This result demonstrated that changing the partner receptors of TEM complexes can target EVs to specific organs of interest.

Integrins are another protein family in EVs, comprising a wide range of cell surface receptors, whose biological functions include regulation of apoptosis, cell proliferation, immune response, and signaling pathways [115,116,117]. They are transmembrane proteins consisting of α and β subunits, whose extracellular domains determine ligand-binding sites of the heterodimers [118]. Integrins play an important role in cancer metastasis, and the different combination of α and β subunits elicit specific targeting of various cell lines and organs [97]. EVs isolated from cancer cell lines containing integrins α6, β4, and β1 subunits prefer to accumulate in the lung. EVs with integrins α6 and β5 subunits were found in Kupffer cells in the liver, while EVs with integrins β3 were detected in the CD31-positive endothelial cells of the brain after administration of EVs into a nude mouse model [97]. EVs from cancer cells or cancer-associated fibroblasts (CAFs) can transform normal fibroblasts in certain organs to create pre-metastatic niches for cancer metastasis through integrins [119]. A study on the mechanism of cancer metastasis through EVs found that EVs from CAFs containing integrins α2β1 could promote lung metastasis in both in vitro and in vivo cancer models [119]. In the study, the lung cancer cell line SACC-LM, fibroblasts isolated from tumor cell lines (CAF-1 and CAF-2), fibroblasts isolated from lung tissues (LFs), and fibroblasts from gingival tissues (NFs) were used for the in vitro assessment of the biological characteristics of EVs, and nude mice were used for the in vivo model [119]. EVs obtained from both the cancer cell lines and CAFs exhibited higher accumulation in the lung but not in the liver and the brain regardless of routes of administration [119]. Notably, EVs from CAFs were more potent in metastasis promotion than EVs from the cancer cell lines, as EVs from CAFs, but not from the cancer cells or from NFs, could be taken up by and subsequently activated in normal lung fibroblasts. EVs from CAFs were grafted with integrin α2β1, but not integrin α4 or α6, thus rendering it specificity against lung fibroblasts [119].

Proteoglycans also contribute to EV targeting and uptake by recipient cells. A study on bovine milk-derived EV uptake found that the glycans on membrane proteins of EVs determine their absorption on human intestinal cells and in mouse models [120]. Among the studied residues, galactose and *N*-acetylgalactosamine residues on EV proteins and *N*-acetylglucosamine residues on proteins of intestinal cells were important for EV uptake, as the removal of these glycans by enzymes or by knockout genes completely blocked EV uptake through the intestine and EV accumulation in the liver and pancreas [120]. EVs that had undergone desialylation were in general taken up more readily compared to normal EVs, but the increase varied between cell lines [121]. Williams and colleagues reported that EVs from the murine hepatic cell line AML12 that had undergone either *N*-glycan removal or sialic acid removal treatment exhibited increased internalization by the skin, lung, retinal, ovarian, and prostate cell lines. In another murine hepatic cell line MLP29, desialylation reduced EV uptake when incubated with National Cancer Institute (NCI) lung cell lines [121]. Two breast cancer cell lines also showed different behaviors against glycosidase-treated EVs even though both cell lines originated from triple-negative breast cancer, indicating that glycans could be used to alter EV uptake by recipient cells [121]. Glycans, together with proteins and lipids, can be utilized for the manipulation of EV biodistribution and uptake with high efficiency and high specificity.

### 3.3. EV Nucleic Acids

Diverse nucleic acid species have been recovered from EVs, including DNA, mRNA, and non-coding RNA. Both genomic DNA fragments and mitochondrial DNA are found in EVs [122,123]. Vagner and colleagues demonstrated that large EVs carry most of the DNA content as compared to small EVs, even though small EVs prevail in terms of quantity [122]. Another study showed that the addition of DNA-harmful reagents to cell culture media increased the amount of DNA in EVs [90]. When combined, similar investigations implied that chromosomal DNA fragments may be enriched in apoptotic bodies as compared to other EV subtypes [124,125,126]. However, in a recent study, Jeppesen and colleagues showed that dsDNA and DNA-binding histones are not carried by small EVs, as the extracellular DNA is susceptible to DNAses, which implied that the DNA is usually co-purified with EVs using conventional methods [66]. With regard to mRNA, Valadi and colleagues demonstrated that EVs carry mRNA contents from donor cells and more importantly these EV-enclosed mRNA contents can be translated into a functional protein upon uptake by the recipient cells [127], which later sparked the researchers’ interest in delivering mRNA to recipient cells via EVs. Further to this, Skog and colleagues introduced Gaussia luciferase (Gluc) mRNA in EVs by lentivirally transducing the donor glioblastoma cells with a vector containing Gluc sequence and showed that human brain microvascular endothelial cells displayed Gluc activity after co-incubation with the EVs containing Gluc mRNA [128]. We also showed that electroporation can introduce mRNAs into EVs for a later functional translation in the recipient cells [49]. Non-coding RNA found in EVs include small non-coding RNA, such as small nucleolar RNAs [129], microRNAs [130], and piwi-interacting RNAs [131], and other long non-coding RNAs [132]. The most studied among them are microRNAs, given the well-known regulatory role of microRNAs and their abundance in EVs [133]. EVs-associated microRNAs function through post-transcriptional regulation of genes in the recipient cells upon uptake, which has been documented in cancers [133], viral infections [134], diabetes [135], and other diseases. Researchers are now considering to use the differential profiles of RNAs, especially microRNAs, in EVs as novel disease biomarkers [136]. The combination of high-throughput methods such as sequencing and highly sensitive methods such as quantitative PCR is commonly used in discovery, validation, and translation [137]. In addition to the protection against enzymatic degradation provided by the EVs’ lipid bilayer, the ability of EV-enclosed RNAs to escape endosomal/lysosomal degradation prompts researchers to utilize EVs as a delivery platform of therapeutic RNAs.

#### Facilitation of Endosomal Escape

Typically, following endocytosis by cells, the endocytosed contents are passed on from the early endosome to the late endosome where the lower pH initiates degradation. Following incorporation in lysosomes, the endocytosed contents are broken down into their constituent components, preventing any endocytosed therapeutic molecules from having any effect on the recipient cell. Endocytosed EVs also follow this pathway of internalization but have shown the ability to transfer their cargo to the cytoplasm through an as of yet unknown mechanism of endosomal escape. This ability of EVs to successfully transfer bioactive cargos to recipient cells is essential for intercellular communication through EVs. Furthermore, many studies have demonstrated that the transfer of RNAs or proteins through EVs results in functional changes in the recipient cell, indicating that not only are the payloads delivered to recipient cells, but they are also able to function and colocalize with necessary elements in the cell to affect the cell’s metabolism [138,139].

Many studies have also shown the ability of EVs to transfer exogenously loaded functional payloads including ncRNAs, mRNA, and plasmid DNA [49,140]. Our study using RBC-derived EVs showed the ability to deliver miR-125b antisense oligonucleotides (ASOs) to target cells, inducing suppression of miR-125b and increased mRNA levels of *BAK1*, the downstream target of miR-125b. We also showed the ability to deliver functional Cas9 mRNA along with a 125b-gRNA, which when administered together were capable of inducing a ~98% decrease in miR-125b and a threefold increase in *BAK1* [49]. This form of endosomal escape has not been shown to have any adverse effects on the cell, such as toxicity or induction of apoptosis, as is the case for many DNA polyplexes such as polyethyleneimine (PEI) polyplexes which make use of the proton sponge mechanism for endosomal escape [141]. Following endosomal escape, PEI was shown to induce cytotoxicity via pore formation in the outer mitochondrial membrane, leading to the release of pro-apoptotic cytochrome C to the cytoplasm, resulting in cell death [142]. In this regard, EVs provide a significant advantage by facilitating the transfer of intact bioactive therapeutics to the cytoplasm where they can achieve their therapeutic effect with lower induction of toxicity compared to chemical-based methods.

### 3.4. EVs as a Whole with Reduced Immunogenicity

While the idea of using nanoparticles for drug carriers is not new, immunogenicity remains a challenge for the therapeutic application of nanoparticles as drug carriers. Many nanoparticles have been generated and modified to circumvent this issue, although the risk of immunotoxicity still exists. For example, mesoporous silica nanoparticles are considered safe because they do not elicit an immune response from lymphoid cells in vitro [143], but when the nanoparticles were injected into mice, a drastic change in spleen weight, splenocyte proliferation, and IgG/IgM levels was observed [144]. Other silica-based nanoparticles also showed increased activation of immune response in in vivo models [145,146]. Second-generation liposomes, the artificial vesicles consisting of at least one lipid bilayer and possessing a long in vivo half-life [147,148], still require surface modifications to enhance their therapeutic window [149]. Even though careful design and in vitro screening already proved engineered liposomes to be safe, repeated injection of liposomes into mice can still generate unwanted immune responses. In a study on RGD-grafted liposomes, Wang and colleagues showed that the repeated injection of engineered liposomes could induce an acute immune reaction in mice [150]. The liposomes contained poly(ethylene glycol) (PEG) and cyclized RGD peptide ligands and were intended for the delivery of cytotoxic drugs targeting the tumor [150]. However, when engineered liposomes were re-administered to the mice, immune responses involving IgG/IgM production, cytokines level elevation, and complement system activation were observed. Lesions were found in the liver, lung, and kidney of the mice, which in turn led to hypothermia and death [150]. In order to reduce anti-PEG IgM production, PEGylated liposomes are either coated with polyglycerol-derived lipids or are modified with the insertion of ganglioside into the liposomes’ lipid bilayers [151,152]. Another strategy to reduce the immunogenicity of PEGylated liposomes is to administer a placebo into the host body at the beginning of the treatment cycles [153]. In the case of PEGylated liposomal doxorubicin, pre-injection with placebo liposomes can reduce the induction of complement activation-related pseudoallergy in the subsequent administrations of drug-carrying nanoparticles [153].

Being the naturally derived vesicles secreted by cells, EVs carry many features from parental cells in terms of lipid, protein, and nucleic acid content, with differences attributed to the enrichment of various groups to suit the EVs’ functions [154]. As EVs carry specific biomarkers from their parental cells, EVs are regarded as safe for use within the same individual or species. But there are considerations regarding the safety of EVs for cross species application. The ability of EVs from exogenous sources to cause immune reactions in the recipient body makes EVs a potential candidate for cell-free vaccines [155], but it may lead to adverse effects if used as drugs or drug carriers. The safety of EVs for cross species application was evaluated in a study based on the administration of HEK-293T-derived EVs in C57BL/6 mice [156]. The study showed that the repeated injection of EVs at 8.5 µg proteins/dose for 10 doses did not elicit a strong immune response in mice. Cytokines IP-10, MDC, and MIP-1β were down-regulated while *VCAM-1* was over expressed, but the weight and histology of major organs such as lung, liver, spleen, thymus, kidneys, heart, and brain were not different compared to the control group [156]. Injection or oral application of bovine milk-derived EVs on C3H/HeNCrl mice also showed the safety of EVs for cross species application [157]. Mice were injected with bovine milk-derived EVs at the dose of 6 mg/kg four times during a 14-day period and were checked for elevation or suppression of cytokines and the presence of a severe allergic reaction. No immunotoxicity was observed in both injected mice and orally fed mice, except for the elevation of IL-6 and G-CSF at 3 h after administration [157]. It is worth noticing that these studies were conducted on a small and homogeneous population of mice. Given that allergy to bovine milk exists within the human population, the usage of EVs from bovine milk, and from other sources in general, should be carefully tested before treatment on a case-by-case basis. This suggests that EVs from selected exogenous sources such as certain mammalian cells could also be used safely for nanomedicine, after extensive characterization.

## 4. Isolation Methods for EV Production

Several techniques have been used for the isolation of EVs from different sources. EVs are isolated based on their density, size, surface biomarkers, lipid composition, or affinity to specific molecules. As EVs are heterogeneous, different isolation techniques and variations in equipment settings would enrich certain EVs subtypes while depleting others [158]. The level of EVs purity also varies, thus giving rise to difficulties in experiment reproducibility and quality control [159,160,161].

Centrifugation is used for the separation of EVs from other contaminants and cell debris in the medium. Briefly, dead cells and large cell debris in cell supernatants or body fluids are removed by successive centrifugations at increasing speeds. EVs are collected from the resulting supernatant and concentrated by ultracentrifugation at speeds ranging from 100,000–200,000× *g* [162,163]. To shorten the procedure, filtration can be applied to remove large cell debris before ultracentrifugation [162]. In some applications that require EVs of higher purity, a 30–60% sucrose cushion or iodixanol can be used in the final ultracentrifugation step to remove vesicle-free proteins or protein-RNA aggregates [162,164,165]. The centrifugation speed and other parameters can be altered based on the EV sources, as the buoyancy and sedimentation efficiency of EVs change with the viscosity of the samples [162]. Since standard protocols for the removal of cell debris and contaminants in different biological fluids have not been defined, EV pools generated by different laboratories would contain biological or chemical variations thus adding confusion to EV research. Shear force is also a factor needed to be considered, as EVs can be altered or destroyed due to the shear force generated during the isolation process [161].

EVs can also be separated based on size. Size exclusion chromatography (SEC) and filtration are two of the most common techniques utilized in this category. In SEC, chromatography columns are packed with porous particles with a defined pore size. The samples are loaded on the surface of packed columns and usually eluted by the same solution as that used for sample dilution (usually PBS). As the samples pass through the column, large particles cannot penetrate the pores and flow out in the void volume, whereas small particles which can freely enter and exit the pores are eluted at a later time [166]. EVs isolated from SEC have sizes greater than 70 nm in diameter [167] and are purified with less than 5% cellular protein contamination [168]. In filtration methods, three sequential filtration steps are used for EV isolation. Cells, cell debris, and EVs larger than 500 nm in diameter are initially removed by normal filtration steps. Following this, proteins that are not associated with EVs and molecules smaller than 10 nm in diameter are removed by tangential flow filtration. EVs are also concentrated at this step. Finally, EVs are sorted into fractions (50, 80, 100, or 200 nm in diameter) by track-etched membranes with various pore sizes [169]. Microfiltration is preferred in large scale production due to its scalable and time-saving procedure while obtaining an EV pool with similar physicochemical characteristics as compared to other methods [170].

Affinity-based purification is another method for the quick isolation of EVs containing specific features or certain groups of surface proteins. Many ligands on EVs have been discovered, which in turn has led to a wide range of available capture molecules for affinity purification. Examples include anti-MHC antigen antibodies [171], anti-heat shock protein antibodies [172], anti-CD9, -CD63, -CD81 antibodies [173], annexin-V, which binds to phosphatidylserine on EVs membrane [174], heparin [175], and lectins, which bind to carbohydrate motifs of glycolipid, glycoproteins, and proteoglycans [176,177,178]. Capture molecules are either used directly, or immobilized on magnetic beads to serve as precipitants, followed by centrifugation for isolation of EV subpopulations [171,172,173], or bound on chromatography gel to serve as the stationary phase in affinity chromatography [179]. The samples are first centrifuged to remove cells and large cell debris. Then, capture molecules or magnetic beads are added to samples and incubated for immunoprecipitation to take place. EVs with specific molecules on the surface will bind to capture molecules with high affinity, which in turn can be precipitated out of the solution. The precipitates containing EVs are centrifuged and washed several times before they are ready for downstream analysis [173]. To avoid physical stress from shear force, EVs can be affinity-captured by column chromatography using a capture molecule-grafted gel. In brief, the samples are pre-treated to remove large cell debris and dead cells and are subsequently loaded on to a chromatography column packed with affinity gel. EVs exhibiting high affinity to the gel are captured and kept inside the column while other unbound molecules are washed out. EVs are eluted by a mobile phase containing a high concentration of salt and are exchanged into corresponding buffers for further analysis [179]. EVs obtained from affinity-based methods are more homologous in terms of specific surface molecules [172,175,179].

EVs can also be precipitated out of solution by addition of polyethylene glycol [180], protamine [181], or sodium acetate [182]. These chemicals change the EV solubility or surface charge, forcing them to aggregate. In the precipitation method, clear supernatants are mixed with aggregating agents and are incubated on ice or overnight at 4 °C to facilitate precipitation. After the aggregates are formed, the samples are centrifuged at 1500–5000× *g*, and the pellets are collected and solubilized in appropriate buffers to obtain the EVs of interest [180,181,182]. This method is quick and simple. However, isolated EVs are heterogeneous, and large protein complexes may be co-precipitated with EVs [180,183].

## 5. Engineering EVs for Drug Delivery

Despite the endogenous advantages of natural EVs for the delivery of bioactive molecules, their usage as a therapeutic vector is limited by three main factors: (1) limited targeting specificity—an efficient drug delivery system requires a versatile method of targeting any desired cell types or organs with high specificity, while maintaining minimal levels of non-specific uptake; (2) poor circulation kinetics—despite their non-immunogenic nature, EVs tend to be cleared from the circulation shortly after systemic administration, decreasing their bioavailability and therapeutic efficacy; and (3) limited therapeutic content—while natural EVs contain some degree of endogenous bioactive molecules, their concentration and efficacy is generally insufficient for therapeutic use. In addition, endogenous bioactive molecules have limited applications from a therapeutic point of view.

Many studies have been conducted in recent years in an effort to overcome the limitations of natural EVs. Much of the studies have focused on the surface modification of EVs to confer greater target specificity and decrease non-specific interactions [140,184,185]. By decreasing non-specific uptake and clearance by cells of the mononuclear phagocyte system, the circulation half-life and bioavailability of EVs can be significantly improved [186,187]. Lastly, novel methods for the efficient loading of EVs have been developed to load a range of different therapeutic payloads into EVs at high concentrations, providing EVs with a greater therapeutic potential [188,189,190]. When utilized together, these modifications can make EVs superior and versatile delivery vectors, capable of targeted delivery of bioactive molecules to target cells, while maintaining the endogenous advantageous characteristics that make EVs desirable as a drug delivery system (Figure 2).

### 5.1. Enhancing Target Specificity

Despite the endogenous homing capability of EVs, this alone is insufficient to achieve targeted delivery of payloads. The natural biodistribution of EVs in vivo is generally divided between many organs, with the homing properties resulting in some organs retaining higher percentages than others [191]. This preferential accumulation is not enough for therapeutic use, as it would lead to a lot of side effects and wastage of the drug due to its uptake by other tissues. To prevent the development of side effects in other tissues and decrease the required dose, the EVs should be able to be delivered specifically to the target site with very high accumulation. This usually requires dedicated functionalization of the EV membrane, allowing the EVs to be taken up efficiently by the target tissue. There are two approaches commonly used to confer targeting capability to EVs: (1) the introduction of targeting moieties such as single domain antibodies, single chain variable fragments (scFv), monoclonal antibodies or targeting peptides that can bind to antigens on target cells and (2) the incorporation of ligands on EVs that can be recognized by specific receptors on the target cells [140,192]. The ligands can be proteins or glycan groups and they facilitate binding and subsequent receptor mediated endocytosis by target cells. While targeting of EVs is essential to achieve the full therapeutic potential of EVs, it should however be noted that surface functionalization of EVs involves the addition of exogenous functional groups such as monoclonal antibodies in high copy numbers, which could in turn induce immune responses such as the development of human anti-murine antibody (HAMA) response. Despite these obstacles, numerous advances have been made to improve the targeting specificity of EVs, while maintaining their safety profile and biocompatible characteristics.

There are two main methods used to confer targeting capabilities to EVs. The first method involves modification of the parent cell via the introduction of a plasmid encoding for the targeting molecule by itself or in conjunction with an existing EV surface protein in the form of a fusion protein, followed by EV isolation. Usually the selected protein is one that is found abundantly in the EVs, so as to obtain a high copy number of the targeting molecule. One of the first studies in this area termed exosome display technology involved the fusion of the desired protein of interest with the C1C2 domain of lactadherin, which then localized to exosomes via binding of the C1C2 domain with exosomal lipids [193]. This method was utilized for the generation of EVs with antibodies against tumor biomarkers and termed ExoMAb, extending the potential of exosome display technology [193]. A similar strategy was also used to express anti-EGFR nanobodies fused to glycosylphosphatidylinositol (GPI) anchored signal peptides [192]. However, even with the identification of a highly expressed surface protein on EVs, the copy number is usually quite low and only confers limited targeting ability. In the above study, this resulted in the targeting EVs having increased binding affinity to EGFR+ cells, but no increase in uptake was seen between the control and the targeted EVs.

Several other studies have also taken similar approaches to targeting and have been met with greater success. One such example is the use of neuron-specific RVG peptide fused to an EV membrane protein called Lamp2b [140]. Following intravenous delivery, the EVs were shown to deliver siRNA specifically to target cells in the brain and induce specific gene knockdown. Of great significance is the observation that no non-specific delivery to non-target tissues was observed. Despite this success, a major shortcoming of this method was identified in the form of degradation of the targeting peptides during biogenesis of EVs [194]. This was solved via the addition of glycosylation of Lamp2b proteins at specific sites, preventing the degradation of the fusion protein, without interfering with its targeting capabilities. The resulting EVs showed enhanced targeting ability in nicotinic acetylcholine receptor expressing neuroblastoma cells. The efficiency and versatility of this method was highlighted by another study that used an integrin specific iRGD peptide fused to Lamp2b to obtain dendritic cell derived EVs capable of targeting tumor tissues and inhibiting growth. In addition to targeting molecules, Zhao et al. also showed that by expressing natural ligands to specific receptors on the surface of plasma membrane derived vesicles (PMVs), the PMVs could be targeted to cells expressing the corresponding receptors [195].

The second method involves direct modification of the EV surface after EV isolation to facilitate addition of the targeting molecules and can be done via the formation of covalent bonds or affinity interactions. The formation of covalent bonds is permanent and usually involves a chemical conjugation step of some nature. While this results in stable conjugation of high copy numbers of the targeting moiety, much of the chemical modification steps that have been used to-date have been shown to affect EV surface chemistry. This was demonstrated by Kooijmans et al. where following chemical conjugation of EGFR targeting nanobodies to the EV surface, a significant increase in uptake by target cells was observed [192]. However, it was also demonstrated that chemical conjugation of non-targeting nanobodies on EVs resulted in decreased association and uptake by cells as a result of the changes to the EV surface following the chemical conjugation step. One of the more advantageous methods is copper-catalyzed azide-alkyne cycloaddition, a form of click chemistry that has minimal effect on EV properties while maintaining efficiency and specificity [196]. This strategy was further extended to conjugate tumor targeting peptides on the surface of exosomes functionalized with azide groups, thereby conferring tumor targeting abilities to the resulting EVs [197].

An alternative to the direct chemical modification of EVs outlined above is the fusion of EVs with micelles or liposomes containing embedded targeting moieties [198]. This method was used for post-insertion of EGFR nanobody-PEG lipids from pre-formed micelles into the EVs. The resulting EVs were shown to exhibit increased binding affinity to EGFR expressing cells [184]. However, this process results in EV-liposomal hybrids, which requires further characterization on biocompatibility and immunogenicity. A more versatile approach was presented by Antes et al. in the form of a modular cloaking and surface display platform. In brief, streptavidin was embedded in the membrane of EVs via an anchor. The streptavidin was then coupled with combinations of biotinylated antibodies and tissue homing peptides, allowing targeted delivery of EVs to desired cells [199]. With a dissociation constant in the order of ~10^−14^ mol/L, the streptavidin-biotin interaction is unmatched among affinity interactions in terms of bond strength. This means that molecules coupled in this way are very stable and will not dissociate in the bloodstream. Several other affinity-based strategies have also been developed to confer targeting abilities, albeit with lower dissociation constants. An innovative approach was taken by Li and colleagues who used surface-carboxyl superparamagnetic iron oxide nanoparticles coated with antibodies against A33 [200]. These nanoparticles were then mixed with EVs derived from A33+ LIM1215 cells, which also expressed A33 on the surface. The resulting complexes formed between the exosomes and the nanoparticles were shown to be able to target A33+ colon cancer cells via free antibodies on the nanoparticles [200].

In addition to protein modifications, glycan modifications have also come to attention as being able to affect the targeting ability of EVs to immune cells. This is especially useful in the development of cancer vaccines, where EVs containing endogenous antigens or loaded with exogenous antigens can be delivered to antigen-presenting cells (APCs), generating a potent antigen-specific immune response. Dusoswa and colleagues demonstrated that the removal of sialic acid residues via desialylation prevented the activation of the immune-inhibitory Siglec ligands on dendritic cells [201]. They further revealed that the addition of ligands for dendritic cell-specific intercellular adhesion molecule-3-grabbing non-integrin (DC-SIGN) such as Lewis^Y^ resulted in increased uptake and presentation of EV-associated antigens to both CD4^+^ and CD8^+^ T cells [201].

### 5.2. Improving Circulation Kinetics

One major drawback of EVs as drug carriers is their short circulation half-life, which limits their therapeutic efficacy [187]. EVs are usually cleared from the circulation within a few hours of systemic administration by cells of the reticuloendothelial system macrophages in the liver and the spleen [202]. However, the specific circulation time and route of EV clearance can vary depending on the subtype, source, and downstream modifications [202]. A combination of both natural features and artificial modifications play a role in determining the circulation time of EVs. These factors usually involve the surface chemistry of EVs, including lipid, protein, and glycan composition.

Multiple pathways have been shown to be responsible for the uptake of EVs, depending on route of administration and EV sources. Clearance from the bloodstream is predominantly mediated via uptake by macrophages of the mononuclear phagocyte system [203]. This is supported by many studies that report uptake of intravenously administered EVs from blood predominantly by the liver and spleen. In a study conducted in rats using RBC-derived vesicles, pre-blocking of scavenger receptors with polyinosinic acid (Poly-I) and PS was able to decrease liver uptake by 40% and increase overall circulation time [187]. In particular, PS-dependent uptake of platelet-derived microvesicles by splenocytes was shown to be dependent on lactadherin in a concentration-dependent manner [204]. It was also reported that cells such as endothelial cells are able to uptake endothelial microparticles (EMPs) in an annexin I/phosphatidylserine dependent manner, where knockdown of annexin 1 on EMPs or silencing of PS receptors on target endothelial cells was able to significantly decrease EV uptake [205]. PS expressing EVs from other sources such as platelets have also shown to be internalized by endothelial cells in a Del-1-dependent manner, where Del-1 monoclonal antibodies against Del-1 were able to inhibit EV uptake [206].

Given the relatively poor circulation kinetics of native EVs, several approaches have been taken to improve their half-life in vivo. They usually involve surface functionalization of EVs using naturally derived ligands or artificial compounds known to prolong circulation half-life, with the aim of decreasing non-specific uptake by cells, thereby increasing the window for specific delivery of encapsulated payloads to target cells. To date, a range of different modifications involving protein and glycan manipulation have been shown to be effective in extending EV half-life in vivo. One of the more promising studies that highlight the advantages of EVs over liposomes explains how the presence of endogenous CD47 on the surface of EVs derived from certain cells confers these EVs with greater circulation half-life as opposed to comparable liposome formulations. CD47 interacts with SIRP-α on macrophages, enabling the EVs to evade phagocytosis. A more artificial modification is polyethylene glycol (PEG) functionalization, one of the more commonly employed strategies which involves incorporation of PEG polymers onto the surface of the EVs. PEG has shown the ability to extend the half-life of both liposomes and DNA polyplexes in vivo. Kooijmans and colleagues demonstrated that EVs with postinsertion of nanobody-PEG micelles had longer half-life in circulation than unmodified EVs [184].

Another approach involves the manipulation of the surface glycoproteins of EVs. Surface glycoproteins play a major role in the biodistribution and uptake of EVs and the addition or removal of certain glycan groups can alter the fate of EVs. Removal of sialic acid from EVs via treatment with neuraminidase was shown to alter the biodistribution of EVs, most likely through the increased retention of EVs in the blood 72 h after an intravenous administration [207]. This is in agreement with a report showing that certain sialylated glycoproteins are rapidly cleared by the asialoglycoprotein receptor [208]. Lastly, it was shown that certain chemical conjugation methods that compromise the functionality of exposed primary amines could help in decreasing non-specific interactions. Following chemical conjugation of EVs with a targeting and control nanobody, the EVs conjugated with the control nanobody had decreased interactions and uptake, which was attributed to the compromised amine groups resulting from the conjugation process [192].

An alternative strategy is preconditioning which involves using specific ligands designed to saturate receptors involved in EV clearance from the circulation. Willekens and colleagues demonstrated that preinjection of PS liposomes decreased liver uptake by 40%, which was accompanied by a slower clearance from the circulation. A similar effect was observed by pre-injection of polyinosinic acid (Poly-I) [187]. Watson and colleagues also report the involvement of scavenger receptors in the clearance of EVs by macrophages and monocytes. Subsequent scavenger receptor class A (SR-A) blockade via dextran sulfate decreased liver uptake by ~50% and significantly improved EV retention in plasma [209].

In addition to the strategies highlighted above, a recent study introduced a novel approach to prevent the clearance of nanoparticles in blood, which involves a transient interaction between nanoparticles and circulating erythrocytes [210]. This strategy, termed erythrocyte hitchhiking involves using affinity interactions to immobilize drug loaded nanoparticles on the surface of erythrocytes. Upon systemic delivery, the nanoparticles were protected from clearance and selectively released at target organs downstream of the injection site. A similar strategy could be adopted for EVs, via the use of bispecific antibodies to bind EVs to surface markers on erythrocytes.

### 5.3. Encapsulation of Therapeutic Payloads

While certain EVs such as MSC-derived EVs have been shown to contain endogenous bioactive cargo with therapeutic values, the majority of EVs lack these endogenous advantages. As such, a number of methods have been developed in an effort to encapsulate exogenous therapeutic molecules inside EVs. The resulting EVs, which are loaded with the desired payloads, are then used for therapeutic treatment. Therapeutic payloads can include a wide variety of molecules, including, but not limited to proteins, peptides, chemotherapy drugs, DNA, and RNA (including siRNAs, miRNAs, ASOs). Given the heterogeneity of the different types of cargo, various methodologies have been developed to load EVs. The type of method to be used is usually dependent on the nature of the biomolecules being loaded and is generally transferrable across EVs from different sources and subtypes. As in the case of surface modification, there exist two major strategies for loading EVs, classified here as endogenous loading and exogenous loading.

#### 5.3.1. Endogenous Loading

Endogenous loading involves transfection of the parental cells, either through direct loading of cells with therapeutic molecules (siRNA, miRNA) or by inducing the production of therapeutic molecules inside the parental cells via a vector encoding for therapeutic molecules (shRNA, mRNA, proteins) [211]. The parental cells are usually transfected using existing transfection protocols such as lipofectamine transfection or electroporation prior to EV isolation. The EV biogenesis machinery is then used to load the transfected payloads into EVs, either through preferential loading via interaction with certain proteins, or simply via passive diffusion of a sufficiently high concentration of therapeutic molecules in the cell. This method is preferable since it doesn’t affect the structure and morphology of the resulting EVs, as opposed to many exogenous loading methods. However, there are many drawbacks, primarily related to the loading efficiency. In the case of endogenous loading of therapeutic molecules, only a small fraction of the encapsulated cargo is going to be transferred to EVs, as even in the case of preferential loading, much of the cargo remains in the parent cell. This means that a large amount of the input cargo is wasted. Moreover, the molecules that are preferentially loaded require modifications to allow their selective accumulation into EVs. These modifications can also subsequently decrease their therapeutic efficacy, defeating the purpose of preferential loading. There is also difficulty in reaching high concentrations of payloads in EVs to be sufficient for therapy, especially in the case of larger molecules that cannot be preferentially loaded such as mRNAs.

However, there have been innovative approaches in recent years to improve endogenous loading of EVs. One of the more direct methods involves cellular nanoporation, where a cellular nanoporation (CNP) biochip was used to produce EVs loaded with nucleic acids of interest such as mRNAs and miRNAs [212]. The loading efficiency was demonstrated to be a thousand-fold higher than found in native EVs and was shown to be capable of inducing translation of tumor suppressor genes, resulting in inhibition of tumor growth in vivo. In addition, this method resulted in significantly larger yields of EVs than obtainable by other conventional methods of EV production. A more intricate method to load large amounts of siRNA into EVs was also developed by Reshke and colleagues, where the desired siRNA sequence was integrated into the backbone of a pre-miRNA (pre-miR-451) [213]. In their study, they demonstrated that pre-miR-451 products are selectively enriched in EVs. Thus, by incorporating the desired RNA within the pre-miR-451 backbone, they were able to obtain between 58-fold to 7000-fold enrichment of the siRNA in EVs, depending on the cell type. This subsequently allowed for efficient gene knockdown in target cells with 10-fold less siRNA than would be required by lipid nanoparticles.

#### 5.3.2. Exogenous Loading

Exogenous loading on the other hand involves direct loading of EVs, post-isolation. Given the inability of EVs to actively endocytose molecules and the lack of a cytoskeletal system within EVs, most traditional transfection methods used for cells cannot be adopted for EV loading. Moreover, cationic transfection agents have the disadvantage of forming EV-sized micelles, which are significantly harder to separate from EVs. Given these obstacles, a number of innovative methods have been developed for the loading of EVs. They commonly involve an agent or process that involves transient permeabilization of the EV membrane, thereby facilitating the entry of the molecules inside EVs, including electroporation, saponin permeabilization, extrusion, and freeze-thaw cycles [188]. Alternatively, the cargo may be modified or associated with molecules which promote its ability to pass through the EV membrane. This frequently involves approaches that increase the hydrophobicity of the cargo, promoting its ability to diffuse through the membrane upon coincubation with EVs [214].

Both these strategies have their own associated problems. Inducing pore formation in the EV membrane results in irreversible changes to the structure and contents of the EVs, which may then alter their pharmacodynamic properties. This is especially true in the case of harsh treatments such as electroporation which has been shown to induce aggregation in many studies [215]. In the case of modification of the cargo, these modifications have a chance of detrimentally affecting the functionality of the modified biomolecules. Table 1 summarizes the strategies that have been demonstrated as being capable of loading EVs with different types of therapeutic molecules, including the associated advantages and disadvantages of each method. Table 2 presents the therapeutic applications of EVs using the aforementioned methods in this review, in which the EVs are classified based on their origins. Specific features of each EV subtype such as their biomarkers and their natural recipient organs are also included.

## 6. Challenges and Obstacles for the Application of EVs

Up to date, there are over a hundred EV-related clinical trials. The majority of these trials evaluate the reliability of EVs as early biomarkers to monitor treatment response in patients, and only a few studies explore the therapeutic potentials of EVs. Indeed, several aspects need to be clarified before EV-based therapeutics can successfully be converted into an approved therapeutic product. In general, preclinical studies tend to focus on the therapeutic benefits upon application of EVs, whereas other equally important aspects are omitted. To date, the safety profile of EVs is still poorly investigated, and only few studies elucidated the potential risk of immunogenicity or toxic effects associated to EVs treatment in proper animal models [156,254]. Furthermore, as we discuss below in this review, technical issues, such as lack of quality control, standardized procedures for production, isolation and storage of EVs, or feasibility of large-scale production are also hampering the clinical translation of EVs-based therapeutics. These obstacles could be overcome via data sharing among researchers and by standardizing manufacturing protocols and regulatory factors for EVs production.

### 6.1. Large-Scale EV Production for Application

For EVs to be applied, we must first be able to produce therapeutic EVs in large quantities. MSCs are commonly used in cellular therapy for immunomodulation and regenerative medicine [255]. With over a thousand clinical trials, MSCs are among the most clinically studied experimental cell therapy platform. Over the years, a great deal of evidence demonstrated that the therapeutic effect of MSCs is mostly linked to the release of paracrine mediators and among these, EVs are the most abundant [256]. MSC-derived EVs recapitulate similar or even better biological activity than parental cells [257], and may definitely serve as an alternative to whole cell therapy. Dendritic cells (DCs) represent another example of a widely studied cell therapy platform that did not satisfy the initial expectation. DCs have an intrinsic ability to stimulate immune responses and therefore their application in immunotherapy is straightforward [258]. The major concern regarding MSC-derived or DC-derived EVs as therapeutics is the difficulty to obtain a large amount of EVs from MSCs and DCs at a low cost affordable enough for a large population of patients. Plant and fruits represent easily accessible sources of EVs. However, immunomodulatory and tissue remodeling activities are reported to previously associate with plant-derived EVs and may confine their clinical usage [259]. In this regard other sources of EVs must be taken into consideration. Among these, human blood represents an unlimited, easily accessible, and safe source of EVs. Nakamura et al. demonstrated that serum-derived EVs accelerated cutaneous wound healing in BALB/c mice [260]. Spurred on by this promising data, the researchers aimed to evaluate the effect of autologous EV rich plasma on cutaneous wound healing [261] or for the treatment of chronically inflamed post-surgical temporal bone cavities [262].

### 6.2. EV Consistency Between Batches

Consistency from batch to batch is essential to ensure drug quality. EVs are naturally derived vesicles with their cargos and membrane characteristics closely related to the health of parental cells. For example, different activation pathways of platelet would yield distinctive EVs subtypes with distinguishable sets of cargos and proteins, even though the source of inactivated platelets, the isolation method and the size of EVs remain the same [263]. Protein concentrations of MSC-derived EVs also vary due to their parental cell origin [264], which further increases the difficulty in maintaining consistency between batches. Moreover, primary cells have limited expansion potential and as such require multiple cell banks for the production of a single batch of EVs [265]. Many cell lines for EVs production are adherent cells and are cultured in 10% fetal bovine serum. However, the use of serum for large scale production is discouraged as serum is expensive, contains bovine EVs which may generate artifacts in the resulting EVs, and may carry unwanted contaminants such as viruses or other infectious agents [266]. The quality of serum also varies between different sources, which could affect the cell health and in turn alter the characteristics of the EVs. Moreover, the effects of culture media on EV generation and cargo loading are not well understood, causing difficulties in troubleshooting when discrepancies in quality arise.

To enhance the consistency between batches, immortalized cell lines with defined characteristics that are widely used in biopharmaceutical production are preferred, as they maintain an infinite expansion capacity. The human embryonic kidney 293 (HEK-293) cell line can be used as a viable cell source for EV production as the cell line is derived from a human, it can be cultured at large scale in a bioreactor [209], it can adapt to serum-free culture medium [209] and has been used for large scale manufacturing of biopharmaceutical agents [267]. Furthermore, its EVs have been proved to be safe for use in an in vivo model [156]. Other than HEK-293 cells, a few attempts have been made for large scale production of EVs from mesenchymal stem cells [268]. However, the use of serum has not yet been eliminated and the cost for maintaining stem cell potency remains high. Other sources of EVs that can be used for large-scale production include bovine milk and red blood cells, as both milk and blood are readily available in large quantities.

The key determinant of EV quality consistency in this case involves the isolation methods. Ultracentrifugation and size-exclusion chromatography are the two most accepted methods for EV isolation in research. However, these methods are not feasible on an industrial scale because they are low-throughput, time-consuming, and the risk of introducing variability between batches is very high. For large-scale production, tangential flow filtration, precipitation, and affinity-based purification are preferred [170,269].

### 6.3. EV Quality Control

Evaluation of EV quality after production is another challenge for manufacturing EVs at an industrial scale. As EVs are nanosized, observation under a conventional light microscope is impossible. EVs characterization through biomarkers has been used intensively in research, but there may be certain artefacts after rounds of particle or protein quantification, immuno-staining, and washing steps, which contribute to variations in quality analysis.

Although the International Council for Harmonisation of Technical Requirements for Pharmaceuticals for Human Use (ICH) has not yet published guidelines for EVs, the same principles for evaluation of drug safety and quality could be applied for EV-based drugs, such as purity, host-cell protein contaminations, or degraded products [270]. Given that each isolation method can only remove 95–97% of protein and lipid contaminations [168,180,183], further refinements with multiple purification methods would be required to obtain EVs of high purity. As EVs are usually isolated from sources of animal or human origin, risk of contamination via viral or infectious agents should be assessed. The storage of EVs is also an issue, given that EVs cannot be stored at room temperature or 4 °C for extended periods of time. Long-term storage of EVs requires that they be stored at −80 °C to retain their characteristics and prevent degradation. This makes it difficult to transport and distribute EV-based drugs to hospitals and clinics [271]. This is especially true in more remote areas where there may be difficulties in transporting and storing EV-based drugs till they are administered. Several approaches such as lyophilization are being investigated in an effort to extend the shelf life of EVs and to allow them to be stored at room temperature, thereby making EV-based therapies more accessible and easier to use.

### 6.4. Risk of Undesirable Biomolecular Transfer to Recipient Cells

EVs contain a multitude of molecules including proteins, RNA, and some amount of DNA which originated from the parental cells. The exact contents of each EV depend on the type of parental cells, the presence or absence of artificial stimulation for EV production (via agents like calcium ionophore), and the method of isolation, among other factors. These molecules may be carried over to recipient cells upon delivery of cargo. Many of these molecules have the potential to induce certain changes in the recipient cells such as activation of signaling pathways or gene regulation through endogenous miRNA. While the effects on the recipient cells are negligible for the most part, there are instances where it could lead to adverse side-effects.

Cancer cell-derived EVs have been used in many studies for the delivery of therapeutic molecules, given the ease with which cancer cells can be cultured indefinitely, to obtain the high EV yields needed for therapy. Given the presence of aberrant mutations prevalent in cancer cells, there is a high risk of mutagenic horizontal gene transfer through cancer cell-derived EVs to the recipient cells, leading to complications [272,273]. An improvement is the use of immortalized cell lines which maintain their replicative potential, however, there still exists the risk that any nucleated cell can potentially lead to oncogenic gene transfer with unpredictable side-effects. One of the simplest solutions to this problem is to use EVs derived from enucleated cells such as RBCs or platelets. RBC-derived EVs in particular have shown great promise, given their lack of nuclear material, the ability to be easily obtained in large amounts and the fact that systemic delivery of EVs is in theory equivalent in safety to a simple blood transfusion [49]. However, for the actual translation of EVs for clinical use, therapeutic EVs would need to undergo full proteomic and DNA/RNA analysis to ensure that their contents are not going to have any adverse effects to the recipient cells targeted in the therapy.

## 7. Conclusions

With the increasing popularity of gene therapy via tools such as CRISPR-Cas9 and RNAi, there is great demand for a safe, versatile, and efficient drug delivery system capable of targeted delivery of these therapeutic cargos to diseased cells via systemic administration. EVs hold great promise in this respect, given their ability to efficiently transfer therapeutic molecules to target cells, while maintaining a safe, low-immunogenic profile. Fuelled by the extraordinary therapeutic potential of EVs, extensive research has been carried out in recent years in an effort to further understand the underlying molecular mechanisms associated with EV biogenesis, biochemical properties, uptake, and their roles in intercellular communication. While these studies have provided a great deal of insight into the function of EVs and their molecular profiles, more investigations are still required. Concurrently, researchers are working to improve or modify the endogenous properties of EVs for therapeutic uses. Great strides have been made in recent years in this respect, focusing mainly on increasing the specific uptake of EVs by target cells, improving circulation kinetics, and exploring novel methods to load therapeutic molecules into EVs. Moreover, these studies have shown the ability to target multiple disease models in vivo through the systemic administration of therapeutic EVs. While each of these breakthroughs has brought us a step closer to achieving the goal of an ideal drug delivery system, extensive and thorough investigations are still required to address many concerns in the field, such as sustaining large-scale EV production, ensuring EV batch-to-batch consistency, assessing EV quality properly, and minimizing the risk of potential horizontal gene transfer via EVs, before EVs can be widely and safely used in clinical settings.

## Figures and Tables

**Figure 1 cells-09-02191-f001:**
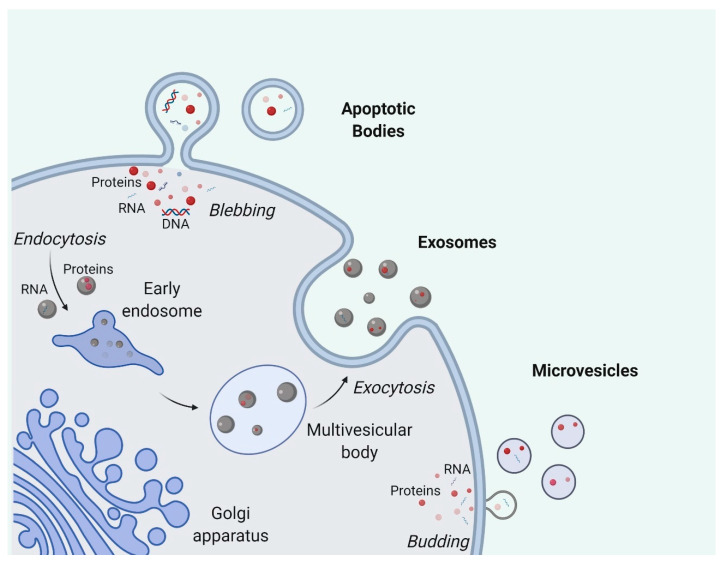
Biogenesis of extracellular vesicle (EV) subtypes, termed exosomes, microvesicles and apoptotic bodies. Exosomes are intraluminal vesicles which are released when a multivesicular body fuses with the cell membrane through exocytosis. Microvesicles are formed by outward shedding of the cell membrane into extracellular space. Apoptotic bodies are generated when cells undergo apoptosis. Figure was generated with BioRender.

**Figure 2 cells-09-02191-f002:**
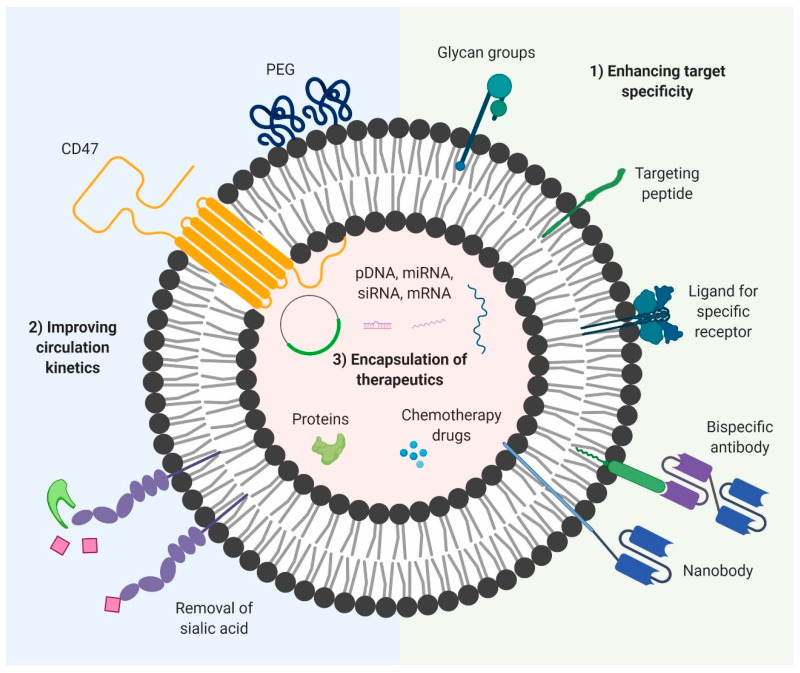
Summary of modifications used to increase therapeutic efficacy of EV-based therapeutics. (**1**) Incorporation of various targeting moieties (nanobodies, bispecific antibodies, peptides) to target specific antigens on target cells or glycan groups/ligands that can bind to receptors on target cells. (**2**) Addition of CD47, PEG or removal of sialic acid to reduce clearance by macrophages of the RES. (**3**) Encapsulation of nucleic acids (plasmid DNA, mRNA, miRNA, siRNA), therapeutic proteins and small molecule drugs which are then delivered to target cells to carry out their therapeutic effects. Figure was created with Biorender.

**Table 1 cells-09-02191-t001:** Strategies used for loading EVs with different types of therapeutic molecules along with associated advantages and disadvantages.

Encapsulation Method	Advantages	Disadvantages	Therapeutic Payloads
Electroporation	Quick and simple	Induces aggregation, alters EV morphology (recovery improved by trehalose pulse media)	Plasmid DNA [189], mRNA, small RNAs (miRNAs, ASOs, siRNAs) [49,140,189,215], Paclitaxel [216], Porphyrins [188]
Sonication	Simple, efficient for small hydrophobic molecules	Cannot load large molecules, EV integrity is compromised	Catalase [217], siRNA [217,218], Paclitaxel [216]
Co-incubation	Simple, doesn’t require specialized equipment	Low efficiency, works only with lipophilic molecules	Paclitaxel [216], hsiRNA [214], Catalase [217], Curcumin [219], Porphyrins [188], Folic acid, Docetaxel [220]
Nanoporation/Cell extrusion	Efficient, can load large charged molecules such as plasmids	Requires parental cell transfection	mRNA [221], siRNA [222]
Endogenous loading of parent cells	Convenient, doesn’t require any treatment of EVs following isolation	Low efficiency (in most cases)	Paclitaxel [223], mRNA [224], miRNA, siRNA [213], TNF-related apoptosis-inducing ligand (TRAIL) [225]
Freeze-thaw method	Simple, doesn’t require specialized equipment	Low efficiency, change in EV size and induction of aggregation	Catalase [217]
Calcium Chloride Transfection	Efficient	Involves introduction of CaCl_2_ precipitate which could introduce toxicity	miRNA mimics/inhibitors [226]
Extrusion	No associated toxicity	Prolonged release of cargo over time	Catalase [217], Porphyrins [188]
Saponin-assisted loading	Efficient for small hydrophilic molecules	Risk of toxicity. Generates transient pores in membrane, cargo may leak out over time. Compromises integrity of EVs	Catalase [217], Porphyrins [188], Doxorubicin [227]
pH-gradient modification	Efficient for small hydrophobic molecules	Damage/denature surface proteins	siRNA, miRNA, ssDNA [228]
Hypotonic dialysis	Improved efficiency for small hydrophobic molecules	Alterations in size and charge of EVs	Porphyrins [188]
Targeted and modular EV loading (TAMEL)	Very high loading capacity	Loaded cargo is rapidly degraded and rendered non-functional	mRNA [229]
Infection of parent cells with viruses	Protects AAV cargo from immune system, improved efficacy as compared to AAV only	Involves the use of AAV viruses which introduces the risk of genotoxicity	Viral capsids (AAV vectors) [230]

**Table 2 cells-09-02191-t002:** An analysis of the different sources of EVs, their specific markers and therapeutic applications. Images of cells were created with Biorender.

EV Source	Recipient Cells/Organs of Unmodified EVs	Markers	Modification	Application	References
Cancer cell lines, immortalized Cell lines 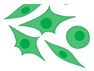	Neuro2a	Liver, Spleen, GI tract, Kidneys	ALIX, TSG101 and CD9	Expression of a fusion protein coding for a nanobody against EGFR	Target tumor cells expressing EGFR	[191,231]
MCF7	CD9, CD81, Rab-5b, CD63, actin, integrin beta 1, HSP70	Encapsulation of Doxorubicin	Breast cancer treatment	[191,231]
HEK-293T	CD81	Expression of DARPin G3 on EV surface	Target HER2+ breast cancer cells	[202,232]
Mesenchymal stem cells 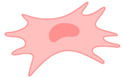	Liver, spleen, lung	CD44, CD63, CD73, CD81, CD90, CD105, CD166	Encapsulation of Paclitaxel	Delivery of chemotherapy for Leukemia therapy	[223,233,234,235,236,237,238]
Encapsulation of miRNA	Increase sensitivity of cancer cells to chemotherapeutics, prevent cancer progression and migration
Dendritic cells 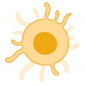	Liver, spleen, lung, GI tractheart, brain, tumor	Lamp-1, Lamp2b, FLOT1, HSP70 (HSPA8, HSP1A1)	Lamp2b fused to neuron-specific RVG peptide	Delivery of siRNA to certain brain regions for specific gene knockdown	[140]
Grafted with DNA aptamer AS1411 targeting nucleolin on breast cancer	Increase accumulation at the tumor	[239]
Encapsulation of Doxorubicin	Targeted delivery of Doxorubicin to cancer cells	[240]
Erythrocytes 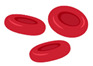	Liver, spleen, bone marrow	Alix, TSG101, CD235a (Glycophorin A), Stomatin	RNA Encapsulation	Delivery of RNA drugs	[49]
Platelets 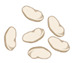	Liver, spleen, lung	CD31, CD41, CD42a, P-selectin,PF4, GPIIb/IIIa, CD9, CD63, TSG101, ALIX	Unmodified EVs	Treatment of cardiovascular disease, enhanced wound healing	[241,242,243]
Fetal Bovine Serum (FBS) 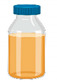	High non-specific uptake	TSG101, CD63, CD81	α-d-mannose with a DSPE-PEG linker	Efficient delivery of immune stimulators and antigens to lymph nodes (dendritic cells),	[244,245]
Bovine Milk 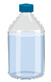	Intestinal mucosa liver, spleen, kidney, lung	Rab-1b, Rab-11a, annexinsCD63, CD59, CD81, miR-30a, miR-223, miR-92a, beta-casein, beta-lactoglobulin mRNA	Encapsulation of chemotherapeutic agents (Paclitaxel, Docetaxel)	Delivery vector for chemotherapeutic agents for cancer therapy	[220,246,247,248]
Encapsulation of Folic acid	Lung and breast cancer reduction
Bacteria 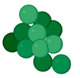	Gram-negative	Liver, lung, spleen, kidney, serum	Porins (Omps, PorA, PorB, and OprF), multidrug efflux pumps, ABC transporters, mobility-related proteins (FliC, PilQ)	Unmodified EVs	Acellular vaccine against *Bordetella pertussis* infection	[155,249,250]
Gram-positive	β-Lactamase, coagulation factor, penicillin binding protein	[251]
Myxobacteria	Immune cells	Chaperonin GroEL1, GroEL2, hydrolase, peptidase	Unmodified EVs	Antibacterial effect against *Staphylococcus aureus*	[252,253]

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
