# Peer review of "Extracellular Vesicles as an Efficient and Versatile System for Drug Delivery"

_cells, 2020, doi:10.3390/cells9102191_

Round 1
Reviewer 1 Report
The review entitled Extracellular Vesicles as an Efficient and Versatile System for Drug Delivery authored by Dang and collaborators is an extensive work comprising the most important aspects of the classification, methods of isolation and potential use of EVs as bioactive cargo delivery system, as well as a good review on the main problems EVs present in terms of large-scale production.
I would like the authors to please address the following points:
- Page 1, line 15: you say one of the main reasons for the failure of novel therapeutics to be clinically translated is the lack of an efficient delivery system. I know this is the abstract, but do you have a reference for this affirmation?
- Page 1, line 40: Could you please mention some of the “safety concerns” of the nanoparticle-based therapeutics?
- Page 1, line 44: The sentence stating a definition for the EVs is very confusing, giving the impression they could even be live organisms. Please rewrite this, explaining they are particles. It is also important to cite that bacteria, mycobacteria, and myxobacteria can release EVs.
- Page 2: lines 66-69: that is a nice explanation of the classical classification of EVs regarding their size and biogenesis. Even though you mention on page 3, lines 121-122, that there are novel definitions of EVs, it would also be nice if you could reference this paper https://doi.org/10.1016/j.cell.2019.02.029 by Jeppesen et al. which also has novel definitions for the EVs regarding their surface markers and biogenesis.
- Figure 1, page 3, line 71: please make the legend more detailed, explaining each type of EV on the image.
- Page 4, line 135: I would replace the word “inborn” for “natural”, “inherent” or “intrinsic”.
- Page 4, line 145: saying that they are biocompatible without considering their specific origin and intended use sounds too optimistic here.
- Page 6, line 233-235: I got confused with this sentence. Do you mean the different isolation methods would give different resulting EVs from the same origin?
- Page 7, line 299: Here you could also discuss the findings of Jeppesen et al.
- Page 8, line 348: Here, again, being too optimistic about the possible biocompatibility or toxicity of EVs. In this scenario, I toxic effect could be concentration-dependent, for example. Please rephrase this.
- Page 8, line 358-359: Immordino et al. (https://www.ncbi.nlm.nih.gov/pmc/articles/PMC2426795/) review liposomes which do not require “rigorous modifications in order to reduce immunogenicity”. I would rewrite this whole sentence with more references. You cannot make such negative affirmations about liposomal formulations when there are several which passed clinical trials and are available on the market (e.g. Doxil, Ambisome). On page 11, lines 491-495 you even mention EVs need surface modification to improve the specificity, just like liposomes.
- Page 9, lines 385-387: “…EVs are a safe choice for nanomedicine, even if EVs are isolated from other species”. We do not know about that yet. Goes et al. (https://www.mdpi.com/2073-4409/9/1/194) isolated EVs from bacteria and one of them showed a cytotoxic effect and induced cytokine release in PBMCs. Again, even when they have a natural origin, toxic effects should not be overlooked. It would also be interesting to mention outer membrane vesicles produced by myxobacteria and their potential as drug delivery systems (https://www.frontiersin.org/articles/10.3389/fmicb.2014.00474/full).
- Page 15: this section about the encapsulation of therapeutic payloads is very nicely written, but please split it in two subsections: endogenous loading and exogenous loading. This will facilitate the reading process.
- Table 2: There are no icons for the FBS and the bovine milk.
- Page 19, section 6.1: please summarize the current alternatives for large-scale EV production.
Author Response
Thank you very much for reading our manuscript and providing your comments. We have edited the manuscript accordingly. Please see the attachment for point-by-point responses to your comments.

Reviewer 2 Report
In this review, the authors focused on the therapeutic potential of EVs as an efficient drug delivery system. The authors performed this review in three parts very good documented. In the first part, they reviewed the biogenesis and biochemical properties of EVs and discussed the current methodologies for their isolation and characterization. The second part is dedicated to describe how EVs can be engineered to further their therapeutic potential and achieve efficient, targeted delivery of encapsulated payloads. In the third part, they outlined the future potential of EV-based therapeutics and discussed the current challenges and obstacles facing their clinical translation.
Minor concerns:
1-On page 5, line 182 is misspelled contribututor, you must type contributor.
2-On page 5, line 183 is misspelled A, you must write As.
Author Response
Thank you very much for reading our manuscript and providing your comments. We have edited the manuscript accordingly.
Comment 1: On page 5, line 182 is misspelled contribututor, you must type contributor.
Response 1: Thank you for pointing this out. We have changed the word to “contributor”.
Point 2: On page 5, line 183 is misspelled A, you must write As.
Response 2: We agree with this and has changed the word to “As”.
Round 2
Reviewer 1 Report
In their revised version, the authors have addressed all my previous comments adequately.
This manuscript is a resubmission of an earlier submission. The following is a list of the peer review reports and author responses from that submission.